# *Anabasis articulata* (Forssk.) Moq: A Good Source of Phytochemicals with Antibacterial, Antioxidant, and Antidiabetic Potential

**DOI:** 10.3390/molecules27113526

**Published:** 2022-05-30

**Authors:** Fakhria A. Al-Joufi, Marwa Jan, Muhammad Zahoor, Nausheen Nazir, Sumaira Naz, Muhammad Talha, Abdul Sadiq, Asif Nawaz, Farhat Ali Khan

**Affiliations:** 1Department of Pharmacology, College of Pharmacy, Jouf University, Sakaka 72341, Aljouf, Saudi Arabia; faaljoufi@ju.edu.sa; 2Department of Biochemistry, University of Malakand, Chakdara Dir Lower 18800, KPK, Pakistan; jmarwa084@gmail.com (M.J.); nausheen.nazir@uom.edu.pk (N.N.); sumaira.biochem@gmail.com (S.N.); livingontheedge36837@gmail.com (M.T.); 3Department of Pharmacy, University of Malakand, Chakdara Dir Lower 18800, KPK, Pakistan; sadiquom@yahoo.com (A.S.); asifnawaz2445@gmail.com (A.N.); 4Department of Pharmacy, Shaheed Benazir Bhutto University, Sheringal Dir Upper 18050, KPK, Pakistan; farhatkhan2k9@yahoo.com

**Keywords:** diabetes, antibacterial activity, total phenolic contents, total flavonoid contents, DPPH, ABTS, antidiabetic enzymes, HPLC-UV, GC-MS

## Abstract

*Anabasis articulata* is medicinally used to treat various diseases. In this study, *A. articulata* was initially subjected to extraction, and the resultant extracts were then evaluated for their antimicrobial, antioxidant, and antidiabetic potentials. After obtaining the methanolic extract, it was subjected to a silica gel column for separation, and fractions were collected at equal intervals. Out of the obtained fractions (most rich in bioactive compounds confirmed through HPLC), designated as A, B, C, and D as well hexane fraction, were subjected to GC-MS analysis, and a number of valuable bioactive compounds were identified from the chromatograms. The preliminary phytochemical tests were positive for the extracts where fraction A exhibited the highest total phenolic and flavonoid contents. The hexane fraction as antimicrobial agent was the most potent, followed by the crude extract, fraction A, and fraction D. DPPH and ABTS assays were used to estimate the free radical scavenging potential of the extracts. Fraction C was found to contain potent inhibitors of both the tested radicals, followed by fraction D. The potential antidiabetic extracts were determined using α-glucosidase and amylase as probe enzymes. The former was inhibited by crude extract, hexane, and A, B, C and D fractions to the extent of 85.32 ± 0.20, 61.14 ± 0.49, 62.15 ± 0.84, 78.51 ± 0.45, 72.57 ± 0.92 and 70.61 ± 0.91%, respectively, at the highest tested concentration of 1000 µg/mL with their IC_50_ values 32, 180, 200, 60, 120 and 140 µg/mL correspondingly, whereas α-amylase was inhibited to the extent of 83.98 ± 0.21, 58.14 ± 0.75, 59.34 ± 0.89, 81.32 ± 0.09, 74.52 ± 0.13 and 72.51 ± 0.02% (IC_50_ values; 34, 220, 240, 58, 180, and 200 µg/mL, respectively). The observed biological potentials might be due to high phenolic and flavonoid content as detected in the extracts. The *A. articulata* might thus be considered an efficient therapeutic candidate and could further be investigated for other biological potentials along with the isolation of pure responsible ingredients.

## 1. Introduction

Humans have employed medicinal plants not only for therapeutic purposes but also for other applications. Since the beginning of human life on earth, they have constantly been utilized by humans for medicinal purposes, which is thus considered the beginning of the exploration of plants for medicinal purposes. Medicinal plants are valuable sources of biodiversity for humanity in providing multiple bioactive secondary metabolites such as sterols, saponins, tri-terpenes, alkaloids, polyphenols, flavonoids, tannins, and essential oils [1]. The mentioned phytochemicals cause different physiological and therapeutic effects when utilized by a human. These effects are broadly summed up as antioxidant, antimicrobial, anti-constitutive, anti-plasmodial, antidiabetic, spasmolytic, and neuroprotective potentials [2,3,4].

Free radicals are chemical species with unpaired electrons that are capable of attacking other chemical substances, especially those containing double bonds. The oxygen and nitrogen-based free radicals are constantly produced in the human body, which can attack biologically important substances, such as DNA and protein, causing a number of health complications from aging to life-threatening cancer and diabetes mellites. Oxidative stress is a general term used to describe such health complications, whereas antioxidants are chemical substances capable of scavenging the responsible free radicals. Most antioxidants contain a benzene ring, which can delocalize the free electrons associated with free radicals. Most of the plant’s secondary metabolites fall into these categories, especially flavonoids and phenolics [5].

Diabetes mellitus is one of the top 10 causative factors of human deaths globally. In individuals with diabetes, there is either less/no production of insulin (type-1) or resistance to the reception of insulin by its receptors (type-2) [6,7]. Type 2 is more prevalent and generally appears as a result of a combination of resistance to insulin action along with an inadequate insulin secretory response [6,7,8]. Although a number of therapies are used to control this dreadful disease, a 100% efficient therapy is still not available. Scientists around the globe are constantly exploring plants for their antidiabetic potentials, and few of them have produced far-reaching results. Extensive research in this regard is still needed as, according to the world health organization, 1.6 million deaths occurred due to diabetes mellitus in 2016. For the treatment and management of this disorder, either insulin is taken or other strategies collectively known as non-insulin treatment are followed. In the non-insulin treatment category, the most popular approach used is to inhibit the carbohydrate metabolic enzymes (α-amylase and α-glucosidase), thus resulting in the minimum release of glucose molecules into the bloodstream [9,10]. Several synthetic inhibitors are commercially available that are taken orally by patients, and although effective, they are associated with more side effects as compared to natural products [11]. Importantly, the trend of pharmacological screening for hypoglycemic and antidiabetic potential has increased manyfold in the last few decades [11]. For ages, medicinal plants have been used as anti-hyperglycemic agents in folk medicine [12,13,14].

*Anabasis articulata* (*A. articulata*; Figure 1) belongs to the genus *Anabasis*, and the family Amaranthaceae is a medicinally valuable plant that is subjected to scientific exploration very little and needs to be explored in line with modern approaches. It is a xerophyte primarily found in deserts. In many parts of the world, it is used in folk medicine to treat skin conditions such as eczema and other ailments, including diabetes, headache, and fever [8].

As mentioned above, *A. articulata* has been reported to have medicinal properties by very few researchers, and thus, the present study is an attempt to explore *A. articulata* for its antimicrobial, antioxidant, and antidiabetic potential in connection to its phytochemical composition, which was investigated using preliminary phytochemical tests, HPLC and GC-MS analysis.

## 2. Materials and Methods

### 2.1. Plant Material Collection

The leaves and stem (1 kg; equal proportions) of *A. articulata* were collected from Mohmand Agency (34.5356° N, 71.2874° E, a deserted tribal area), Pakistan, in the year 2020–2021. The collected plant samples were identified by the Taxonomist, working as an expert in Herbarium, University of Malakand, Pakistan. A voucher specimen (BGH-UOM-190) was kept for record in the herbarium. After cleaning, the samples were shade dried, grounded, and then subjected to the extraction process.

### 2.2. Chemicals and Reagents

All chemicals used in the study were of analytical grade except those used as solvents in HPLC analysis. The reagents; 2,2-Diphenyle-1 picrylhydrazyl (DPPH), sodium carbonate, Folin-Ciocalteu (F-C) reagent were obtained from Sigma-Aldrich CHEMIE GmbH, St. Louis, MO, USA, whereas quercetin and 2,2′-Azino-Bis-3 ethyl benzothiazoline-6-sulfonic Acid (ABTS) aluminum chloride, methanol, sodium hydroxide, ethanol, sodium nitrite, and ascorbic acid were purchased from Sigma-Aldrich, Taufkirchen, Germany.

### 2.3. Extraction and Fractionation 

Extraction and fractionation were carried out according to already reported protocols [6,13]. About 200 g of the powdered sample was soaked in methanol, filtered through the Whatman filter paper, and concentrated using a rotary evaporator (Rotavapor R-200, Buchi, Flawil, Switzerland). About 20 g of crude extract was obtained and used in subsequent experiments. The extract was eluted through a silica gel column for fractionation as per the following details: The extract was mixed with silica gel slurry and then allowed to dry in the air. The sample loaded silica was then carefully loaded to a large silica gel column with an internal diameter of 10 cm and packed height of 50 cm using a gradient of increasing polarity from *n*-hexane to ethyl acetate as the mobile phase. The oil fraction was extracted through *n*-hexane solvent. The effluents of columns were separated into four purified fractions designated as A, B, C, and D. Fraction A was separated by silica gel column chromatography in a solvent system of ethyl acetate and *n*-hexane (5:95), fraction B was separated by silica gel column chromatography in a solvent system of ethyl acetate and *n*-hexane (10:90), fraction C was separated in a solvent system of ethyl acetate and *n*-hexane (20:80), while fraction D was separated by silica gel column chromatography in a solvent system of ethyl acetate and *n*-hexane (30:70). The resultant extracts were stored at a temperature of 4 °C in a refrigerator till their screening for in vitro biological activities.

### 2.4. Preliminary Phytochemical Analysis 

Reported protocols were followed for the identification of major phytochemicals in the extracts [15].

### 2.5. Estimation of Total Phenolic Content (TPC) and Total Flavonoid Content (TFC)

Shirazi et al.’s method was used for the estimation of TPC [16] in extract, whereas Kim et al.’s method [17] was followed for TFC estimation. From the stock sample (5 mg/5 mL), 1 mL was added to 9 mL of distilled water, to which 1 mL F-C reagent was added and incubated for 6 min, after which 10 mL of 7% sodium carbonate solution and 25 mL of distilled water were added. The absorbance was measured at 760 nm after 90 min of incubation time. The standard gallic acid solution’s curve was used for the estimation of the TPC expressed as mg GAE (Gallic acid equivalent)/g of the dry sample.

In distilled water (500 µL), 100 µL of each sample, 150 µL of aluminum chloride, 100 µL of 5% sodium nitrate, and 200 µL of 1M sodium hydroxide were added for assessment of TFC. The absorbance of the mixture was recorded at 510 nm after being incubated for 5 min. The TFC was calculated as mg QE (quercetin equivalents)/g of the dry sample.

### 2.6. HPLC-UV Characterization

Methanolic extract, hexane, and purified fraction were each added to a distilled water:methanol (1:1) mixture, heated at 50 °C for 1 h, dually filtered, and poured into HPLC vials [18]. The extract’s phytochemicals were separated using HPLC Agilent 1260 with an eclipsed C18 column (Santa Clara, CA, USA). The spectrums were recorded at 320 nm. Retention times of the available standards were employed to identify the unknown compounds present in the analyzed samples. Quantification of antioxidants was measured by formula (Equation (1)):(1)Cx=Ax×CsμgmL×VmLAs×Sample wt. in g
where *Cx* = concentration of unknown sample, *As* = peak area of standard, *Ax* = peak area of unknown sample, *Cs* = concentration of standard (0.09 µg/mL).

### 2.7. GC-MS Analysis

The extracts were further analyzed for volatile components using GC-MS (Agilent Technological USA) analysis [19]. The mass spectra and retention time of the compounds present in samples were compared with those of Willy and NIST libraries [20].

### 2.8. Antibacterial Screening

The antibacterial potential of methanolic crude extract, purified fractions (A, B, C, and D), and oil fraction was assessed against *Shigella dysentery* (*S. dynasties*), *Escherichia coli* (*E. coli*), and *Salmonella Typhi* (*S. Typhi*) using the agar disc diffusion method. The strains were grown on nutrient broth, whereas the antibacterial spectrum of the extracts and fractions was assessed by the agar disc diffusion method, as mentioned before. A control (ampicillin) disc was also placed. The Clinical Laboratory Standards Institute guidelines (CLSI 2012) were followed [3] while determining the zone of inhibition (ZI) encountered by each extract and fractions against selected bacterial strains. The activity for each extract, purified fractions, and oil fraction was performed in triplicate and presented as the mean values.

### 2.9. Antioxidant Activities

#### 2.9.1. DPPH Assay

A slightly modified DPPH assay as used before by Brand William was followed [21]. The absorbance of 3 mL from stock DPPH (20 mg in 100 mL of methanol) was adjusted to 0.75 at 517 nm. The DPPH stock solution was covered and kept in the dark overnight to generate free radicals. About 2 mL of each dilution (1000, 500, 250, 125, 62.5 µg/mL) prepared from methanolic extract stock (5 mg/5 mL of methanol) were mixed with 2 mL of pre-incubated DPPH stock solution and incubated for 15 min. Ascorbic acid was used as a standard. The absorbance of the reaction mixtures was recorded at 517 nm and the % inhibition was calculated as (Equation (2)):(2)$$\%\mathrm{inhibition}=\frac{A-B}{A}\times 100$$
where A = absorbance of pure DPPH in oxidized form, B = absorbance of the sample, which was measured after 15 min of reaction with DPPH.

#### 2.9.2. ABTS Assay

A standard protocol for ABTS free radical scavenging potential of extracts was followed [22]. ABTS (7 mM) and K_2_S_2_O_8_ (2.45 mM) were mixed (in methanol) and put in the dark for 24 h for free radicals’ formation, which was then used as stock solution. The absorbance of 3 mL from the stock ABTS was adjusted to 0.75 at 745 nm, which was considered a control. About 300 μl of each of the serial dilutions of methanolic extract (1000, 500, 250, 125, 62.5 µg/mL) and 3 mL of stock ABTS were mixed and incubated for 15 min at 25 °C, and their absorbance was measured at 745 nm. Ascorbic acid was used as a control. The scavenging activity was calculated by Equation (2).

### 2.10. In Vitro Antidiabetic Activities

#### 2.10.1. Inhibition of α-Amylase

The extracts were assessed as inhibitors of α-amylase following a standard protocol reported in the literature with some modifications [23]. In distilled water, an alpha-amylase stock solution (10 mg/100 mL) was prepared. About 10 µL of alpha-amylase stock solution was mixed with 30 µL of each sample dilution and 40 mL of starch solution, and these were kept at 37 °C for 30 min. After incubation, 20 µL of HCl (1M) was added to the reaction mixture, and its absorbance was measured at 580 nm. Acarbose was used as the reference standard. Alpha-amylase % inhibition was calculated as (Equation (3)):(3)$$\%\alpha -\mathrm{amylase}\;\mathrm{inhibition}=\frac{\mathrm{control}\;\mathrm{absorbance}-\mathrm{sample}\;\mathrm{absorbance}}{\mathrm{control}\;\mathrm{absorbance}}\times 100$$

#### 2.10.2. Inhibition of α-Glucosidase

The reported protocol was followed to assess the α-glucosidase inhibitory potential of the extracts [24]. To 100 µL α-glucosidase (0.5 units/mL), 50 µL of each sample dilutions and 600 µL of 0.1 M phosphate buffer (pH 6.9) were mixed. From the substrate (*p*-nitrophenyl-α-D-glucopyranoside) prepared as a 5 Mm solution in 0.1 M phosphate buffer, 100 µL of the substrate was added to the reaction mixture and kept for 15 min at 37 °C. The absorbance was recorded at 405 nm. The reaction mixture without α-glucosidase was labeled as blank, whereas the reaction mixture without the sample was taken as the control. The degree of the enzyme’s activity inhibition was measured as:(4)$$\%\alpha -\mathrm{glucosidase}\;\mathrm{inhibition}=\frac{\mathrm{control}\;\mathrm{absorbance}-\mathrm{sample}\;\mathrm{absorbance}}{\mathrm{control}\;\mathrm{absorbance}}\times 100$$

### 2.11. Statistical Analysis

All in vitro experiments were performed in three replicates. All results have been presented as mean ± SEM. The Student’s *t*-test and one-way ANOVA followed by Dunnett’s post hoc multiple comparison test was used to evaluate the significance of the data obtained. *p* ≤ 0.05 was considered significant.

## 3. Results

### 3.1. Yield from Fractionation 

Crude extract: 10 g, fraction A: 200 mg, B: 150 mg, C: 120 mg, and D: 100 mg were obtained. About 80 mg of purified hexane fraction was obtained in semi-solid form. In the subsequent studies, these components have been tested.

### 3.2. Qualitative Phytochemical Screening 

Plants contain millions of compounds that are classified into roader groups of phytochemicals. Such constituents are determined as preliminary evaluations to decide the medicinal value of a plant. Table 1 represents the presence of different phytochemical groups showing that this plant is worthy of being investigated for its medicinal profile, being a rich source of phytochemicals.

### 3.3. TPC and TFC in Resultant Extracts

A standard Gallic acid curve was constructed by preparing the dilutions 20, 40, 60, 80 and 100 mg/mL to estimate the TPC in different tested samples of *A. articulata* using a graphical regression method (Figure 2A). Comparatively higher TPC contents were estimated in almost all extracts as compared to TFC. Results show that the highest TPC (Figure 3) values were observed for crude extract and then oil and fraction A (72.1 ± 0.2, 69.0 ± 1.1 and 68.0 ± 0.4 mg GAE/g of dry sample, respectively).

To estimate the TFC in different tested samples of *A. articulata*, a regression curve of standard quercetin was constructed by preparing the dilutions 20, 40, 60, 80 and 100 mg/mL. The estimated contents are graphically presented in Figure 2B. Fraction A/crude extract followed by the oil fraction has shown the highest total flavonoid contents (62.0 ± 0.1, 62.3 ± 1.2, and 55.1 ± 0.3 mg QE/g of dry sample, respectively).

### 3.4. HPLC-UV Analysis

The HPLC chromatograms of the crude extract are presented in Figure 4A, purified fractions in Figure 4B–E, and the *n*-hexane fraction in Figure 4F. The compounds identified were: malic acid, gallic acid, chlorogenic acid, epigallocatechin gallate, Bis-HHDP-hex (pedunculagin), morin, 3-0-caffcoylquinic acid, Ellagic acid, kaempferol-3-(p-coumaroyl-diglucoside)-7-glucoside, catechin hydrate, rutin, syringic acid, quercetin-7-O-sophoroside, kaempferol-3-(caffeoyl-diglucoside)-7-rhamnosyl, mannose, pyrogallol, caffeic acid, mandelic acid, quercetin-3-(caffeoyldiglucoside)-7-glucoside, p-Coumaric acid, galactose, vitamin C, 3-0-caffeoylquinic acid, apigenin-7-O-rutinoside, quercetin 3,7-di-O-glucoside, rhamnose, 3,5-dicaffeoylquinic acid, mandelic acid, xylulose, quercetin 3,7-O-glucoside, glucose, quercitin-3-0-glysides, quercitin-3-O-rutinoside, quercitin-3-O-glycosides, quercetin, quercetin-3-(caffeoyldiglucoside)7-glucoside, quercitin-3-0-glycosides, in crude extract, various fraction (A, B, C, D, and *n*-hexane fraction. Each peak in the given chromatograms represents a phytoconstituent. For the identification of such constituent retention times of each component were compared with that of external standards. The quantification of each phenolic compound with their particular peak position and retention time (Rt) in chromatogram is presented in Table 2, Table 3, Table 4, Table 5, Table 6 and Table 7.

### 3.5. GC-MS Characterization of the Different Fractions

#### 3.5.1. Purified Fraction A

The GC-MS chromatogram of fraction A is indicated in Figure 5A. Appendix A represents the structural formulas of five phytochemical compounds identified from the given chromatogram, whereas Appendix A represents their mass fragmentation pattern and Appendix A represents different parameters of the major phytochemical compounds identified.

#### 3.5.2. Purified Fraction B

The GC-MS chromatogram of fraction B is indicated in Figure 5B. The technique confirmed the presence of 12 phytochemical compounds in fraction B, and their other parameters are presented in Appendix A. Appendix A represent the structural formulas of 12 phytochemical compounds and the pattern of their mass fragmentation, respectively.

#### 3.5.3. Purified Fraction C

Figure 5C shows the GC-MS chromatogram of fraction C. Appendix A represents 13 phytochemical compounds identified along with some basic parameters related to the analysis performed. Appendix A indicates the structural formulas of 13 phytochemical compounds, and Appendix A indicates their mass fragmentation pattern.

#### 3.5.4. Purified Fraction D

The GC-MS chromatogram of fraction D is presented in Figure 5D, where the presence of 13 phytochemical compounds was confirmed as presented in Appendix A along with analysis-related parameters. Appendix A represents the structural formulas of phytochemical compounds, whereas Appendix A represents the pattern of their mass fragmentation.

#### 3.5.5. Oil Fraction

Figure 5E represents the GC-MS chromatogram of the purified oil fraction. Figure 5F shows the structural formulas of major phytochemical compounds (a and b), and Appendix A shows their different parameters.

### 3.6. Antibacterial Activity

The results of the antibacterial potential of the samples have been tabulated in Appendix A and graphically represented in Figure 6. The results depicted that all the samples except for B and C showed activity against the tested bacterial strains. The broad-spectrum antibiotic ampicillin was used as a positive control. The *n*-hexane (oil) fraction showed the highest ZI against all tested strains: *S. dysentery*, *E. coli*, and *S. Typhi* as 20, 24, and 16 mm respectively. An appreciable degree of antibacterial bacterial potential suggests the plant’s possible usage as a source for isolating antibacterial compounds.

### 3.7. Antioxidant Activity of Crude and Purified Fractions of A. articulata

#### 3.7.1. DPPH Assay

Almost all extracts inhibited DPPH free radicals; however, among them, *n*-hexane fraction, crude extract, and fraction B showed significant free radical inhibition with IC_50_ values of 45, 90, and 62 µg/mL, respectively, as presented in Appendix A and Figure 7A.

#### 3.7.2. ABTS Assay

The ABTS scavenging potential of extracts is presented in Appendix A and Figure 7B. The results depict that fraction A and *n*-hexane extract possesses significant free radical inhibition with the lowest IC_50_ values of 75 and 71 µg/mL, respectively, as compared to the ascorbic acid used as the standard, which showed an IC_50_ value of 32 µg/mL.

### 3.8. In Vitro Antidiabetic Activities of Purified Fraction and Extract

#### 3.8.1. In Vitro α-Glucosidase Inhibition

Figure 8A and Appendix A represent the %α-glucosidase inhibition of crude extract, oil, and the purified fractions. The crude extract showed the highest inhibition of the enzyme with an IC_50_ of 32 µg/mL followed by fraction B, which produced an IC_50_ of 60 µg/mL.

#### 3.8.2. In Vitro α Amylase Inhibition

As shown in Figure 8B and Appendix A, the %α-amylase inhibition is appreciable, and the highest inhibition was caused by crude extract with an IC_50_ of 34 µg/mL followed by fraction B, which produced an IC_50_ of 58 µg/mL.

## 4. Discussion

Presently, insulin therapies are the treatment of choice to control hyperglycemia in diabetes mellitus. Other strategies are the inhibition of alpha-amylase and glucosidase through different inhibitors, as both enzymes are responsible for releasing glucose from starch taken in food [28,29]. In this context, an attempt has been made in this study to identify the possible antidiabetic phytochemical that could inhibit the activity of carbohydrate digesting enzymes (α-amylase and α-glucosidase). The study revealed that the plant could be a potential candidate for isolating antidiabetic compounds.

With the increasing reports about the side effects of synthetic drugs, researchers have focused on plants to isolate effective therapeutic precursors with low or no side effects. Drug resistance is the other overwhelming problem in the modern era, and the search for new antibiotics of plant origin is in progress. The plant crude extract and purified fractions showed appreciable antibacterial activity, which is evident from the zones of inhibitions against selected bacterial strains, as shown in Figure 5.

Oxidative stress caused by free radicals that are constantly produced during normal metabolic processes is a serious health threat. Although these are constantly deactivated by the human defense system, in the modern era, humans have started relying on processed foods, which have given rise to the overproduction of free radicals. Research shows that plants could neutralize the free radicals because of their constituent phenolics [30], as benzene rings in such compounds can stabilize the singlet electron of the free radicals. Collectively, such phytoconstituents are named antioxidant compounds, which play an important role in human health by combating reactive oxygen species and, in turn, is the main contributor to a number of human diseases, including insulin resistance, cardiovascular diseases, atherosclerosis, and coronary heart disease. Butylated hydroxytoluene and butylated hydroxy anisole are strong synthetic antioxidant agents, but they are carcinogenic and toxic to humans. Therefore, plant-based phenolic compounds can be used as antioxidants to scavenge or stabilize free radicals involved in oxidative stress generated in human bodies as a result of oxidation of certain substances. It is found that the use of synthetic antioxidants is injurious to human health, and individuals taking them are at risk of cancer and other liver disorders. The antioxidants in plants have become a hotspot for researchers in recent decades due to the mentioned fact of low or no side effects. Studies have indicated that the use of natural antioxidants can reduce oxidative stress and reduce the risk of major human diseases, including oxidative stress [3,6]. The *n*-hexane fraction, crude extract, and fraction B were more potent against DPPH radicals, whereas against ABTS, the *n*-hexane fraction and fraction A were more potent, indicating that these extracts contained certain phytoconstituents capable of scavenging free radicals, which could thus be further investigated for the isolation of responsible compounds. The DPPH radicals in acholic medium undergo a reduction in the presence of hydrogen donating antioxidants. Phytochemicals such as flavonoids and phenolics are good antioxidants and play a vital role in scavenging the free radicals due to the presence of benzene rings in their structures [6,7,8,9,10,11,12,13]. The observed antioxidant potential can be correlated with the estimated TFC and TPC values, as these are the responsible scavengers in the extracts. The total polyphenol and flavonoid content in the fractions increased in the following order: crude extract, fraction A, and oil fraction. The crude extract has the highest TPC and TFC, i.e., (TPC = 72.1 mg GAE/g and TFC = 62 mg QE/g) followed by purified fraction A, which has the highest TPC and TFC, i.e., 68 GAE/g and 62 mg QE/g, inferring the plant is a good source of flavonoids and phenolics. As mentioned before, due to the presence of benzene rings in the structure, flavonoids and phenolics have been found to be excellent scavengers of the free radicals, which is why the tested radicals, ABTS and DPPH, were potently scavenged by the extracts, i.e., the total phenolic and flavonoid contents in the extracts and purified fractions were positively proportional to the antioxidant activities. The current results were in line with the previously reported studies [6,14]. The study of Kim et al. [31] showed plants that contained high TFC and TPC, and by virtue of these components, they exhibit various biological potentials. Their conclusion was based on findings of extracts from 40 plant species in Korea. As mentioned, phenolic and flavonoid compounds are strong antioxidants that can deactivate free radicals by offering their hydrogen atoms and electrons [32], which is the reason that plants with high TFC and TPC inhibit DPPH and ABTS radicals more potently in laboratory-scale experiments. The positive correlation between the total phenolic content and flavonoid content in the plant extracts and the antioxidant activities have been observed by other researchers as well [32]. The plants in the form of extracts could, therefore, offer strong activity against a wide range of oxidants and thus would have great medicinal applications. It can be seen from Appendix A that the crude extract and fractions exhibited significant activities against the DPPH and ABTS tested radicals, which needs to be further investigated. Furthermore, for the crude extract, the preliminary phytochemical tests (Table 1) were positive, indicating the presence of broad phytochemical groups and, consequently, the wider range of their therapeutic action.

The HPLC analysis of crude and purified fractions of *A. articulata* showed the presence of several possible compounds that might be responsible for antioxidant and antidiabetic activities. The antidiabetic properties of *A. articulata* crude extracts and fractions (Figure 7 and Appendix A) were determined based on the inhibitory effect against two carbohydrate hydrolyzing enzymes, namely α-amylase and α-glucosidase. As mentioned before, starch is converted into disaccharides and oligosaccharides by pancreatic α-amylase, while disaccharides are broken down into glucose by intestinal α-glucosidase [3,6] and, thus, if inhibited, will lessen the glucose burden in diabetic patients as their inhibition could retard the breakdown of starch in the gastrointestinal tract and, therefore, would ameliorate hyperglycemia in human. The detected compounds are known to be antioxidant and antidiabetic agents, as indicated in the previously reported studies [3,6,9,14]. The current results of the screening are in close accordance with the already reported study of Nazir et al., where they confirmed the presence of quercetin, morin, and rutin in the methanolic extract of *Silybum marianum* (L.) seeds [33] and in the methanolic extracts of the fruit of *Elaeagnus umbellata* Thunb. [6]. The results of this study are in agreement with the findings of other studies where strong antioxidant activities were observed along with strong α-glucosidase and α-amylase inhibitions [3,6,9].

The medicinal plant has become a vital source of antioxidants in the last few decades. Literature surveys have shown that the ingestion of natural antioxidants can reduce oxidative stress-related diseases. Various studies have shown that the presence of malic acid, gallic acid, quercetin, morin, ellagic acid, rutin, chlorogenic acid, and epigallocatechin gallate can be liable for the antioxidant capacity observed [14,34,35]. It is evident from the literature that gallic acid, chlorogenic acid, epigallocatechin gallate, and morin have strong antioxidants and antidiabetic potentials [36,37].

The GC-MS analysis of the purified fraction also confirmed the presence of certain valuable phytochemical compounds: Acetdimethylamide, *N*-Nitrosomorpholine, 1,2-Benzenedicarboxylic acid, Mono(2-ethylhexyl) phthalate, Bis(2-ethylhexyl) phthalate, N-Acetyl-l-methioninamide, 2-Propanamine, Phenol, 2,4-di-tert-butyl, Benzene, (1-dodecyltridecyl)-, Benzene, (1-hexyltetradecyl)-, Benzene, (1-hexylheptyl)-, Isopropyl Palmitate, 10-Octadecenoic acid, methyl ester, 1-Docosene, Methyl ricinoleate, Oleic acid, tetradecyl ester, Diisooctyl phthalate, Asparagine, entacosane, 13-phenyl Eicosane, 7-phenyl, Dodecane, 6-phenyl, Palmitic acid, methyl ester, tert-Hexadecanethiol, Decyl oleate, octadecyl ester, Elaidic acid, isopropyl ester, Phenethyl alcohol, á-methyl, Benzyl-3-hydroxypyrrolidine, Diethyl Phthalate, 2,6-Dimethyl-pyridine-3,5-dicarboxylic acid, dihydrazide, Methoxycarbonyl-2-methoxyphenyl isothiocyanate, Phosphoric acid, dibutyl 3-trifluoromethyl-3-pentyl ester, 4-Acetylaminophthalic acid, dimethyl ester, Benzene-1,3-dicarboxylic acid, 5-acetylamino-, (2-Phenyl-1,3-dioxolan-4-yl) methyl (9E)-9-octadecenoate, 1-Heneicosyl formate, 18,19-Secoyohimban-19-oic acid, Cleavamine, 18á-carboxy-3,4à-dihydro-, 1-Piperidinecarboxaldehyde, and (1-Ethyl-propenyl)-dimethyl-amine, which could possibly have their share in the observed biological potentials. The findings of the present study could be correlated with the reported studies [19,38]. From the rich phytochemical composition of the selected plant, we hypothesized that the different levels of antidiabetic activity of the extract and different fractions of *A. articulata* are due to the varying levels of various phytochemicals in each extract/fraction. The purified fraction A followed by crude extract *A. articulata* exhibited higher levels of TPC and TFC, together with antioxidant and antidiabetic activity as compared with the other extracts/fractions. This indicates that phenolic compounds, including flavonoids, are key active compounds found in these extracts, and the plant could thus be a good candidate for further studies to isolate inhibitors of the tested radicals and enzymes.

## 5. Conclusions

From the current study results, it can be concluded that *A. articulata* in extract form should be considered an effective source to relieve oxidative stress and health complications associated with diabetes. This plant also has the potential to be used as an antimicrobial agent as it effectively inhibited the growth of selected bacterial strains. The α-amylase and α-glucosidase enzymes were inhibited by extracts to an appreciable extent suggesting that this plant could be used as a potential candidate to isolate effective antidiabetic drugs. The observed biological activities were at the expense of its rich phytochemical composition, confirmed through HPLC-UV and GC-MS techniques in this study. Further studies are needed to isolate pure compounds responsible for the observed biological potentials.

## Figures and Tables

**Figure 1 molecules-27-03526-f001:**
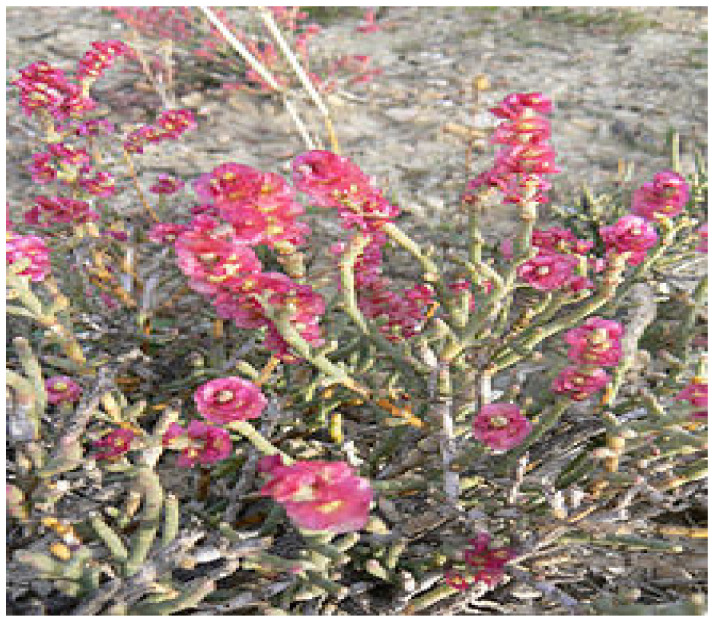
The *A. articulata* plant.

**Figure 2 molecules-27-03526-f002:**
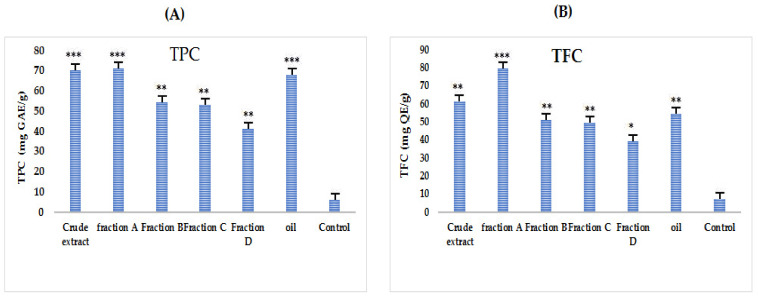
Total phenolic (**A**) and total flavonoids (**B**) contents in different tested samples (crude extract, purified fractions, and oil) of *A. articulata*. (**A**) TPC expressed as gallic acid equivalents (mg GAE)/g dry plant sample; (**B**) TFC expressed as quercetin equivalents (mg QE)/g dry plant sample. The data is represented as Mean ± SEM, *n* = 3. Values are significantly different as compared to positive control * *p* < 0.05, ** *p* < 0.01, *** *p* < 0.001.

**Figure 3 molecules-27-03526-f003:**
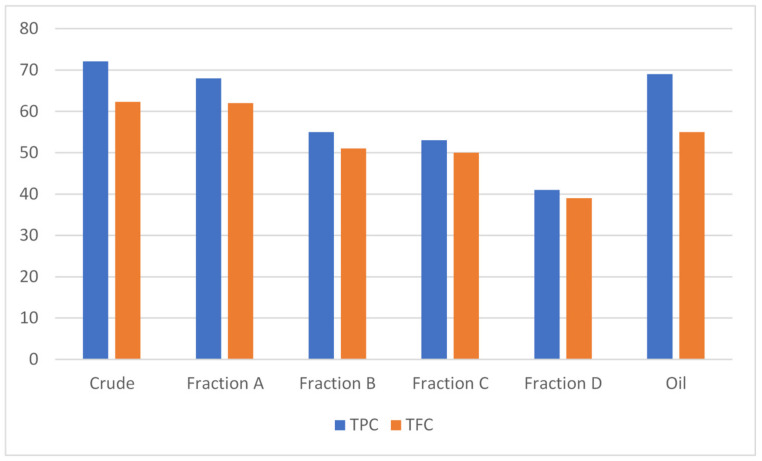
Total phenolic and total flavonoid content in different tested samples (crude extract, purified fractions, and oil) of *A. articulata*. TPC expressed as gallic acid equivalents (mg GAE)/g of dry plant sample, and TFC expressed as quercetin equivalents (mg QE)/g of dry plant sample. The data are represented as mean ± SEM, *n* = 3.

**Figure 4 molecules-27-03526-f004:**
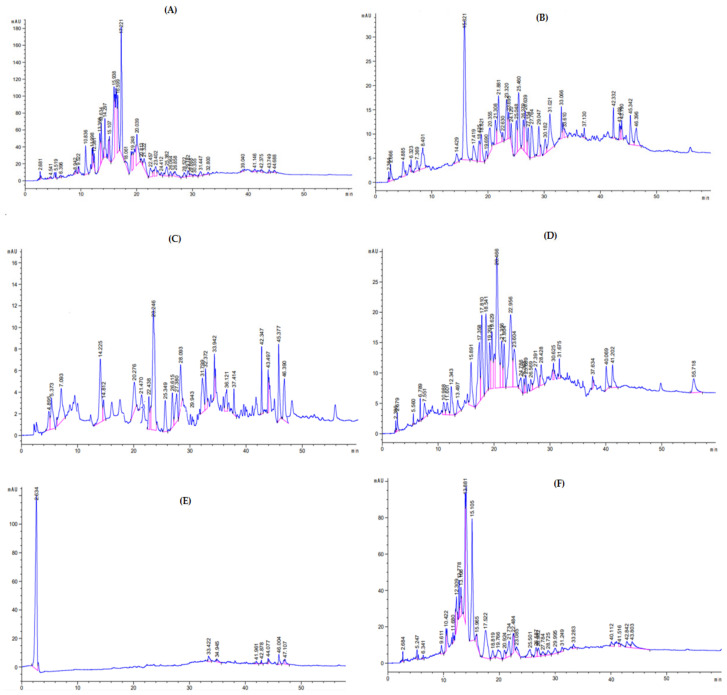
HPLC chromatograms of crude extract (**A**), various purified fractions (**B**–**E**), and *n*-hexane fraction (**F**) of *A. articulata*.

**Figure 5 molecules-27-03526-f005:**
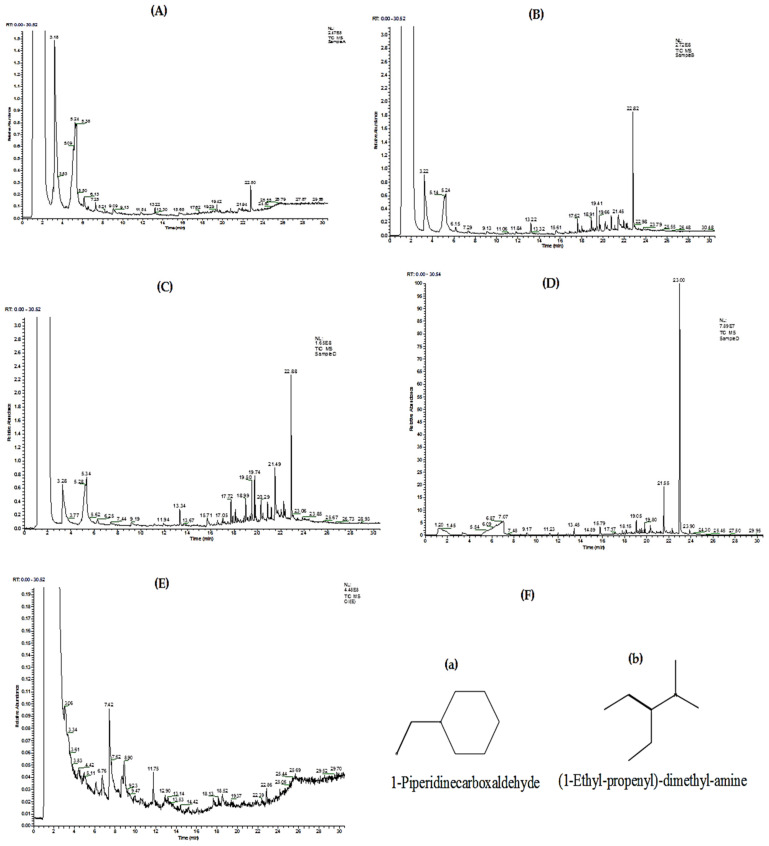
GC-MS chromatogram of *A. articulata* purified fractions (**A**–**D**)**,** oil fraction (**E**), and major phytochemical compounds (**a**,**b**) identified in the oil fraction (**F**).

**Figure 6 molecules-27-03526-f006:**
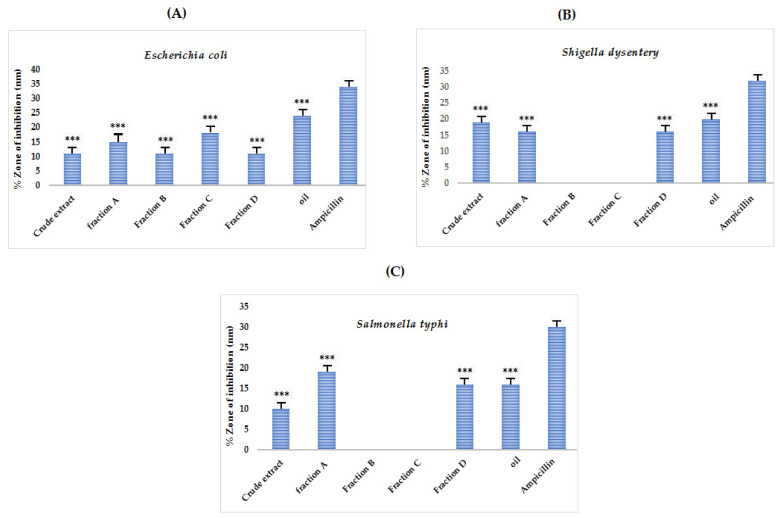
Antibacterial potential of crude extract, purified fractions (A, B, C, and D), and oil fraction of *A. articulata* against (**A**) *E. coli*, (**B**) *Shigella dysentery* and (**C**) *S. typhi*. The data are represented as mean ± SEM, *n* = 3. Values are significantly different as compared to positive control **** p* < 0.001.

**Figure 7 molecules-27-03526-f007:**
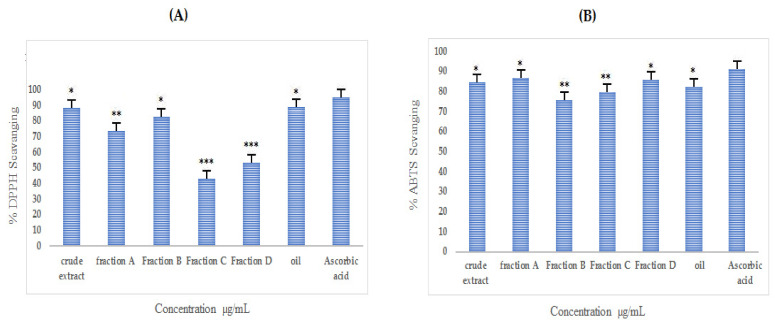
Antioxidant activity ((**A**) = DPPH; (**B**) = ABTS) of crude extract, purified fractions (A, B, C, and D), and oil fraction of *A. articulata*. The data plotted is mean ± SEM, *n* = 3. Values are significantly different as compared to positive control * *p* < 0.05, ** *p* < 0.01, *** *p* < 0.001.

**Figure 8 molecules-27-03526-f008:**
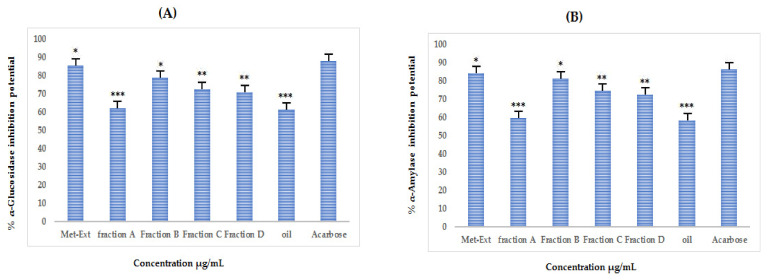
(**A**) %α-glucosidase inhibition potential and (**B**) %α-amylase inhibition potential of crude extract, purified fractions (A, B, C, and D), and oil fraction of *A. articulata* at different concentrations. The data are represented as mean ± SEM, *n* = 3. Values are significantly different as compared to positive control * *p* < 0.05, ** *p* < 0.01, *** *p* < 0.001.

**Table 1 molecules-27-03526-t001:** Phytochemical screening (qualitative) of *A. articulata* crude extract.

Phytochemical	Reagent	Observation	Result
Alkaloids	Dragendroff’s	Reddish-orange precipitate	+
Tannins	Gelatin	Dirty (brownish) green precipitates	+
Flavonoids	Ferric chloride	The yellowish appearance that becomes clear after acid (HCL) addition	+
Triterpenoids	Liebermann Burchard	Brown ring	+
Glycosides	Keller Killiani	Reddish-brown layer	+

**Table 2 molecules-27-03526-t002:** Identified phytochemicals in crude extract of *A. articulata* through the HPLC-UV technique.

Retention Time (min)	Phytochemical Compounds	HPLC-UV λmax (nm)	Peak Area of Sample	Peak Area of Standard	Concentration (µg/mL)	Identification Reference
2	Malic acid	320	53.130	40.323	1.186	Ref. Stand
4	Gallic acid	320	41.239	195.40	0.189	Ref. Stand
6	Chlorogenic acid	320	32.966	12.929	2.295	Ref. Stand
8	Epigallocatechin gallate	320	44.782	7261.474	0.005	Ref. Stand
11	Bis-HHDP-hex(pedunculagin)	320	171.562	-	-	[25]
12	Morin	320	103.604	2.00	46.622	Ref. Stand
14	3-0-caffcoylquinic acid	320	514.593	-	-	[25]
16	Ellagic acid	320	912.321	319.242	2.572	Ref. Stand
18	Kaempferol-3-(p-coumaroyl-diglucoside)-7-glucoside	320	149.535	-	-	[25]
20	Catechin hydrate	320	810.747	78.00	9.355	Ref. Stand
22	Rutin	320	264.573	22.40	10.630	Ref. Stand
23	Syringic acid	320	254.546	-	-	[26]
24	Quercetin-7-O-sophoroside	320	50.8907	-	-	[26]
25	Kaempferol-3-(caffeoyl-diglucoside)-7-rhamnosyl	320	261.997	-	-	[26]
26	Mannose	320	85.1536	-	-	[25]
28	Pyrogallol	320	101.640	1.014	90.213	Ref. Stand
29	Caffeic Acid	320	129.708	-	-	[25]
30	Mandelic acid	320	45.405	72.00	0.567	Ref. Stand
31	Quercetin-3-(caffeoyldiglucoside)-7-glucoside	320	107.633	-	-	[27]
32	p-Coumaric acid	320	52.304	-	-	[27]
42	Galactose	320	57.017	-		[25]

**Table 3 molecules-27-03526-t003:** Identified phytochemical compounds in fraction A of *A. articulata* through the HPLC-UV technique.

Retention Time (min)	Phytochemical Compounds	HPLC-UV λmax (nm)	Peak Area of Sample	Peak Area of Standard	Concentration (µg/mL)	Identification Reference
4	Vitamin C	320	64.43	22.4	2.588	Ref. Stand
6	Chlorogenic acid	320	18.64	2.929	5.727	Ref. Stand
8	Epigallocatechin gallate	320	132.75	7261.474	0.0164	Ref. Stand
14	3-0-caffeoylquinic acid	320	45.345	-	-	[26]
18	Kaempferol-3-(p-coumaroyl-diglucoside)-7-glucoside	320	86.189	-	-	[26]
18	Apigenin-7-O-rutinoside	320	128.126	-	-	[26]
20	Catechin hydrate	320	134.971	78.00	1.557	Ref. Stand
22	Rutin	320	23.593	22.40	0.947	Ref. Stand
23	Syringic acid	320	176.298	-	-	[25]
23	Quercetin 3,7-di-O-glucoside	320	77.612	-	-	[25]
24	Quercetin-7-O-sophoroside	320	58.34	-	-	[25]
25	Kaempferol-3-(caffeoyl-diglucoside)-7-rhamnosyl	320	248.248	-	-	[26]
26	Rhamnose	320	103.713	-	-	[26]
27	3,5-dicaffeoylquinic acid	320	97.727	-	-	[26]
29	Caffeic acid	320	113.173	-	-	[27]
30	Mandellic acid	320	63.944	72.00	0.799	Ref. Stand
31	Quercetin-3-(caffeoyldiglucoside)-7-glucoside	320	223.444	-	-	[26]
42	Galactose	320	60.609	-	-	[27]
43	Xylulose	320	31.328	-	-	[27]

**Table 4 molecules-27-03526-t004:** Identified phytochemicals in purified fraction B of *A. articulata* through the HPLC-UV technique.

Retention Time (min)	Phytochemical Compounds	HPLC-UV λmax (nm)	Peak Area of Sample	Peak Area of Standard	Concentration (µg/mL)	Identification Reference
4	Gallic acid	320	36.4984	195.40	0.168	Ref. Stand
14	3-0-caffeoylquinic acid	320	160.818	-	-	[26]
20	Catechin hydrate	320	78.1380	78.00	0.902	Ref. Stand
22	Rutin	320	42.5292	22.40	1.709	Ref. Stand
23	Syringic acid	320	324.245	-	-	[26]
25	Kaempferol-3-(caffeoyl-diglucoside)-7-rhamnosyl	320	58.9445	-	-	[26]
27	3,5-dicaffeoylquinic acid	320	88.6597	-	-	[26]
28	Pyrogallol	320	85.8767	1.014	76.221	Ref. Stand
31	Quercetin-3-(caffeoyldiglucoside)-7-glucoside	320	88.6597	-	-	[26]
32	p-Coumaric acid	320	56.5793	-	-	[26]
43	Xylulose	320	36.3262	-	-	[27]

**Table 5 molecules-27-03526-t005:** Identified phytochemicals in fraction C of *A. articulata* through the HPLC-UV technique.

Retention Time (min)	Phytochemical Compounds	HPLC-UV λmax (nm)	Peak Area of Sample	Peak Area of Standard	Concentration (µg/mL)	Identification Reference
2	Malic acid	320	24.7858	40.323	0.554	Ref. Stand
11	Bis-HHDP-hex (pedunculagin	320	47.4527	-	-	[25]
12	Morin	320	100.496	2.00	45.223	Ref. Stand
18	Apigenin-7-O-rutinoside	320	246.3633	-	-	[26]
22	Rutin	320	424.706	22.40	17.064	Ref. Stand
23	Quercetin 3,7-O-glucoside	320	201.7868	-	-	[26]
25	Kaempferol-3-(caffeoyl-diglucoside)-7-rhamnosyl	320	30.1412	-	-	[26]
26	Rhamnose	320	32.8371	-	-	[27]
27	3,5-dicaffeoylquinic acid	320	120.040	-	-	[26]
28	Pyrogallol	320	66.255	1.014	58.806	Ref. Stand
30	Mandelic acid	320	29.289	72.00	0.366	Ref. Stand
31	Quercetin-3-(caffeoyldiglucoside)-7-glucoside	320	32.0566	-	-	[26]
37	Glucose	320	21.6425	-	-	[27]
41	Quercitin-3-0-glysides	320	78.9891	-	-	[26]

**Table 6 molecules-27-03526-t006:** Identified phytochemicals in purified fraction D of *A. articulata* through the HPLC-UV technique.

Retention Time (min)	Phytochemical Compounds	HPLC-UV λmax (nm)	Peak Area of Sample	Peak Area of Standard	Concentration (µg/mL)	Identification Reference
2	Malic acid	320	2514.20	40.323	56.116	Ref. Stand
34	Quercitin-3-O-rutinoside	320	41.0693	-	-	[26]
41	Quercitin-3-O-glycosides	320	30.249	-	-	[26]
42	Galactose	320	16.116	-	-	[27]

**Table 7 molecules-27-03526-t007:** Identified phytochemicals in *n*-hexane fraction of *A. articulata* through the HPLC-UV technique.

Retention Time (min)	Phytochemical Compounds	HPLC-UV λmax (nm)	Peak Area of Sample	Peak Area of Standard	Concentration (µg/mL)	Identification Reference
2	Malic acid	320	28.2673	40.323	0.631	Ref. Stand
6	Chlorogenic acid	320	14.9574	2.929	4.595	Ref. Stand
10	Quercetin	320	137.065	90.90	1.357	Ref. Stand
11	Bis-HHDP-hex (pedunculagin)	320	44.509	-	-	[26]
12	Morin	320	231.396	2.00	104.128	Ref. Stand
18	Apigenin-7-O-rutinoside	320	89.384	-	-	[26]
22	Rutin	320	159.626	22.40	6.414	Ref. Stand
23	Syringic acid	320	76.785	-	-	[26]
25	Kaempferol-3-(caffeoyl-diglucoside)-7-rhamnosyl	320	152.318	-	-	[26]
26	Rhamnose	320	100.004	-	-	[26]
31	Quercetin-3-(caffeoyldiglucoside)7-glucoside	320	51.863	-	-	[26]
41	Quercitin-3-0-glycosides	320	31.616	-	-	[26]
42	Galactose	320	100.903	-	-	[27]

## Data Availability

The data presented in this manuscript belong to the research work supervized under Muhammad Zahoor and have not been deposited in any repository yet. However, the data are available to the researchers upon request.

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
