# Peer review of "Anabasis articulata* (Forssk.) Moq: A Good Source of Phytochemicals with Antibacterial, Antioxidant, and Antidiabetic Potential"

_molecules, 2022, doi:10.3390/molecules27113526_

Round 1

Reviewer 1 Report

The manuscript entitled “Anabis Atriculata (Forssk.) Moq: A Good Source of Phytochemical Having Antibacterial, Antioxidant, And Antidiabetic Potential” reports a phytochemical study and antibacterial, antioxidant and Antidiabetic properties of Anabis Atriculata extacts.

I found the research question was clearly stated and well expressed objectives. The methods applied were seems to be appropriate and fully described in the article. Similarly, the results section well furnished, illustrative and self-explanatory enough, to answer the questions posed at the beginning. Considering the applicability and utility of the work, I recommend the acceptance of the paper after minor revision, according the following points:

  1. Some corrections should be made (forgotten points to add or others to delete)? to check.
  2. Page 3, line 117, “…ethyl acetate ethyl acetate….”. Remove the repetition.
  3. In Figure 3, the structures of compounds “a” and “b” are not clear and need to be redrawn.
  4. In section 3.6. Antibacterial activity, I would like to suggest to the authors to express the results of the antibacterial activity in Minimal Inhibitory Concentration (MIC).

Author Response

Reviewer 1

The manuscript entitled “Anabis Atriculata (Forssk.) Moq: A Good Source of Phytochemical Having Antibacterial, Antioxidant, And Antidiabetic Potential” reports a phytochemical study and antibacterial, antioxidant and Antidiabetic properties of Anabis Atriculata extacts.

I found the research question was clearly stated and well expressed objectives. The methods applied were seems to be appropriate and fully described in the article. Similarly, the results section well furnished, illustrative and self-explanatory enough, to answer the questions posed at the beginning. Considering the applicability and utility of the work, I recommend the acceptance of the paper after minor revision, according the following points:

  • Thank you worthy reviewer for the encouraging remarks.
  1. Some corrections should be made (forgotten points to add or others to delete)? to check.
  • The manuscript was thoroughly checked and such types errors were corrected accordingly.
  1. Page 3, line 117, “…ethyl acetate ethyl acetate….”. Remove the repetition.
  • Thank you worthy reviewer, the repetition was deleted accordingly.
  1. In Figure 3, the structures of compounds “a” and “b” are not clear and need to be redrawn.
  • Thank you worthy reviewer, the figure resolution was increased and now the compound structures are clear.
  1. In section 3.6. Antibacterial activity, I would like to suggest to the authors to express the results of the antibacterial activity in Minimal Inhibitory Concentration (MIC).

Thank you worthy reviewer, we have not determined the MIC values of the extracts. If the reviewer still we will perform the additional experiments. 

Reviewer 2 Report

Topic of this study was to obtain the  different fractions from the Anabasis articulata tissue, and next to test their potential biological activity. Modern analytical equipment used by the authors  allow them to identify several phytocompounds known from their pro-healthy properties These preliminary results might useful in the future study on this plant. However, some points for consideration to improve manuscript should be addressed:

1. The authors should provide a bit more information about the tested plant material i.e. how far collected material is representative and reflect the chemical properties of this species; please add some information about the  location - what is the Mohmand agency - it doesn't have to be clear to everyone. How many plants were taken? What was the share of the various parts (leaves:stems) for extraction; Photos of the plants would be appreciated.

2.To see clearer the contribution of the TFC to the TPC, these contents should be shown in the same picture. It seems that in some extracts TFC are similar or even higher (fraction A) than TPC. How does this relate to the chromatographic separation?

Why the concentration of phenolic compounds shown in Figure 1 (TFC/TFC) and Table 2 (chromatographic separation) are different?

3. The discussion is too general. For example, the extended third paragraph adds little in the context of the presented research. Here are rather commonly known facts presented. The authors should pay attention to which compounds are especially interesting, e.g. compounds which distinguishing this  from other medicinal plants (compounds quantitatively dominant in the raw material and / or known to have high biological activity). Since the study concerns only one factor, such comparisons should be more detailed than those presented.

I also have the feeling that there are too many references to authors own research in the discussion, please add other works.

Author Response

Reviewer 2

Topic of this study was to obtain the  different fractions from the Anabasis articulata tissue, and next to test their potential biological activity. Modern analytical equipment used by the authors  allow them to identify several phytocompounds known from their pro-healthy properties These preliminary results might useful in the future study on this plant. However, some points for consideration to improve manuscript should be addressed:

  • Worthy reviewer, thank you for the encouraging remarks.

  1. The authors should provide a bit more information about the tested plant material i.e. how far collected material is representative and reflect the chemical properties of this species; please add some information about the  location - what is the Mohmand agency - it doesn't have to be clear to everyone. How many plants were taken? What was the share of the various parts (leaves:stems) for extraction; Photos of the plants would be appreciated.
  • Worthy reviewer, about one kilogram leaves and stem were dried which were then powdered. About 200 g powder were used for extraction. All these information are highlighted in the revised manuscript. From 200 g powders 20 g crude extract was obtained. The location information was accordingly provided. About 1 kg leaves and stems were used in same proportion (1:1) that resulted into 200 g mass after drying. The plant picture was inserted accordingly.
  1. To see clearer the contribution of the TFC to the TPC, these contents should be shown in the same picture. It seems that in some extracts TFC are similar or even higher (fraction A) than TPC. How does this relate to the chromatographic separation?
  • Worthy reviewer, the figure was accordingly replaced.

Why the concentration of phenolic compounds shown in Figure 1 (TFC/TFC) and Table 2 (chromatographic separation) are different?

  • Worthy reviewer, the sample size are different as these two different type of analysis. HPLC is highly sophisticated machine as compared to a spectrophotometer. Also in figure the TPC and TFC represents group of compounds whereas table 2 representing individual compounds. Also methos of extraction in both cases are different.
  1. The discussion is too general. For example, the extended third paragraph adds little in the context of the presented research. Here are rather commonly known facts presented.The authors should pay attention to which compounds are especially interesting, e.g. compounds which distinguishing this  from other medicinal plants (compounds quantitatively dominant in the raw material and / or known to have high biological activity). Since the study concerns only one factor, such comparisons should be more detailed than those presented.
  • Worthy reviewer, the discussion was improved accordingly. However, as we not isolated individual compounds therefore, at this stage such discussion will not be fruitful.

I also have the feeling that there are too many references to authors own research in the discussion, please add other works.

  • Some the references were replaced by other relevant references.

Reviewer 3 Report

This manuscript describes many active compounds and preliminary test on anti-bacterial, antioxidant and anti-diabtic activities. of A. articulata. It should be investigated for the mechanism of action and their pharmacological avctivities of  individual/pure compound in further study. However, some issues are edited.

For methods, subtitle of  3.5.1-3.5.4 need to make consistency.

For discussion, authour should be mention the association of free radicals and cause of diabetic mellitus.

Author Response

Reviewer 3:

This manuscript describes many active compounds and preliminary test on anti-bacterial, antioxidant and anti-diabtic activities. of A. articulata. It should be investigated for the mechanism of action and their pharmacological avctivities of  individual/pure compound in further study. However, some issues are edited.

For methods, subtitle of  3.5.1-3.5.4 need to make consistency.

  • Thank you worthy reviewer, the consistency was accordingly brought into the headings

For discussion, authour should be mention the association of free radicals and cause of diabetic mellitus.

  • Thank you worthy reviewer, the discussion was accordingly modified.